# Staphylococcal Enterotoxin C2 Mutant-Induced Antitumor Immune Response Is Controlled by CDC42/MLC2-Mediated Tumor Cell Stiffness

**DOI:** 10.3390/ijms241411796

**Published:** 2023-07-22

**Authors:** Xuanhe Fu, Mingkai Xu, Zhixiong Yu, Wu Gu, Zhichun Zhang, Bowen Zhang, Xiujuan Wang, Zhencheng Su, Chenggang Zhang

**Affiliations:** 1Institute of Applied Ecology, Chinese Academy of Sciences, No. 72 Wenhua Road, Shenyang 110016, China; fuxuanhe@163.com (X.F.); guwu20@mails.ucas.edu.cn (W.G.); zhangzhichun19@mails.ucas.edu.cn (Z.Z.); zhangbowen20@mails.ucas.edu.cn (B.Z.); dudu-00200@126.com (X.W.); zhenchengsu@iae.ac.cn (Z.S.); zhangcg@iae.ac.cn (C.Z.); 2Department of Immunology, Shenyang Medical College, No. 146 Huanghe North Street, Shenyang 110034, China; yuzhixiong@symc.edu.cn; 3Key Laboratory of Superantigen Research of Liao Ning Province, Shenyang 110016, China; 4University of Chinese Academy of Sciences, Beijing 100049, China

**Keywords:** staphylococcal enterotoxin C2, cell softness, ovarian cancer

## Abstract

As a biological macromolecule, the superantigen staphylococcal enterotoxin C2 (SEC2) is one of the most potent known T-cell activators, and it induces massive cytotoxic granule production. With this property, SEC2 and its mutants are widely regarded as immunomodulating agents for cancer therapy. In a previous study, we constructed an MHC-II-independent mutant of SEC2, named ST-4, which exhibits enhanced immunocyte stimulation and antitumor activity. However, tumor cells have different degrees of sensitivity to SEC2/ST-4. The mechanisms of immune resistance to SEs in cancer cells have not been investigated. Herein, we show that ST-4 could activate more powerful human lymphocyte granule-based cytotoxicity than SEC2. The results of RNA-seq and atomic force microscopy (AFM) analysis showed that, compared with SKOV3 cells, the softer ES-2 cells could escape from SEC2/ST-4-induced cytotoxic T-cell-mediated apoptosis by regulating cell softness through the CDC42/MLC2 pathway. Conversely, after enhancing the stiffness of cancer cells by a nonmuscle myosin-II-specific inhibitor, SEC2/ST-4 exhibited a significant antitumor effect against ES-2 cells by promoting perforin-dependent apoptosis and the S-phase arrest. Taken together, these data suggest that cell stiffness could be a key factor of resistance to SEs in ovarian cancer, and our findings may provide new insight for SE-based tumor immunotherapy.

## 1. Introduction

Bacterial superantigen staphylococcal enterotoxins (SEs) constitute a class of biological macromolecules with molecular weights in the range of 22–28 kDa, which are produced by Gram-positive Staphylococcus aureus [1]. As potent T-cell activators, SEs can directly bind to the antigenic groove of the major histocompatibility complex class II (MHC II) molecule and the specific Vβ chain of the T-cell receptor (TCR), resulting in hyperactivation of T cells. In this manner, activated T cells release large amounts of proinflammatory cytokines such as interleukin-2, tumor necrosis factor-α, and interferon-γ, which activate the cytotoxic T lymphocytes (CTLs) and nature killer (NK) cells to secrete perforin and granzyme B [2]. Next, perforin forms pores in the membrane of target cells, allowing the granzyme B to enter into the cell and induce cell apoptosis [3]. As a member of the SE family, staphylococcal enterotoxin C2 (SEC2) has been applied in clinical trial as an effective tumor immunotherapeutic agent in China for over twenty years [4]. To enhance antitumor activity, our laboratory constructed an MHC-II-independent mutant of SEC2 named ST-4 with enhanced TCR Vβ recognition [2,5]. However, the antitumor activity induced by SEC2/ST-4 on different cancer cells was significantly different. The mechanisms of immune resistance to SEs in cancer cells is still unclear.

Perforin plays a critical role in cytotoxic effects of CTLs and NK cells [6]. Tumor cells show various escape mechanisms protecting against perforin activity [7]. Recently, biophysical experiments revealed that increasing target cell membrane tension promoted pore formation on cell membrane by perforin, implying that target cell membrane tension controls the cell killing effect of perforin-producing CTLs [8]. In addition, using atomic force microscopy (AFM), Xu et al. found that ovarian cancer cells are generally softer than other types of epithelial cancer cells. Cell stiffness is an important factor of the metastatic potential of ovarian cancer cells [9]. These data strongly suggest that ovarian cancer cells may rely on intrinsic cell softness to prevent perforin-induced killing.

To date, little is known about signaling mechanisms for regulating cell stiffness in ovarian cancer cells. Cell stiffness is regulated by the local interaction of fibrillar actin (F-actin) and interleaved with the molecular motor myosin II [10]. Myosin II is the major motor protein responsible for the generation of membrane tension [11]. Inhibiting myosin II will increase membrane tension [12]. The activity of myosin is controlled by phosphorylation of myosin II light chains (MLC2). As the upstream signaling molecule of MLC2, members of the Rho GTPase have been reported to regulate the kinases responsible for phosphorylating MLC2 [13]. Among the Rho GTPases family, cell division control protein 42 (CDC42) is frequent overexpression in epithelial cancers, especially in ovarian cancers [14,15]. Serine 71 phosphorylation of CDC42 can interact with downstream effector molecules such as CDC42 effector proteins (CDC42EPs) and p21-activated kinase (PAK) that participate in cell shape and cellular motility [16,17,18].

To better understand SEC2/ST-4-induced antitumor immune effect and drug resistance mechanisms in ovarian cancer cells, RNA-seq were performed on the SKOV3 cells and ES-2 cells with or without treatment of SE2/ST-4 to evaluate the changed expression in apoptosis-related gene and signaling pathways. According to the results of the differentially expressed genes (DEGs) associated with the cell cytoskeleton and AFM data, we found that the cell stiffness of ES-2 cells was softer than SKOV3 cells. ES-2 cells might intensify cell softness through the CDC42 signal pathway. Next, we chose siRNA CDC42 and blebbistatin, the myosin II ATPase inhibitor, to investigate the role of the CDC42/PAK/MLC2 signaling pathway in regulating ovarian cancer cell stiffness. Cell growth assays and Western blot analysis revealed that CDC42/PAK/MLC2-regulated cell stiffness plays an important role in the SEC2/ST-4-induced CTL killing of tumor cells. After enhancing ovarian cancer cell stiffness, we found that SEC2/ST-4 could exhibit a significant antitumor effect against SE insensitive ES-2 cells by promoting the perforin-dependent apoptosis pathway. The purpose of this work was to understand SEC2/ST-4-induced antitumor and resistance mechanisms in ovarian cancer cells. Understanding mechanisms of tumor resistance provides important insight into superantigen-based tumor immunotherapy.

## 2. Results

### 2.1. Analysis of SEC2/ST-4-Stimulated PBMC Activation and the Production of Cytotoxic Factors

CFSE-labeled PBMCs were stimulated with SEC2/ST-4 at a final concentration of 100 ng/mL for 72 h, and proliferation was analyzed by flow cytometry. As shown in Figure 1A,B, SEC2 and ST-4 significantly induced PBMC proliferation. At the same concentration, the stimulatory activity of ST-4 was significantly higher than SEC2 (*p* < 0.05).

SEC2/ST-4-induced perforin and granzyme B secretion were detected by ELISA. As shown in Figure 1C,D, SEC2 and ST-4 could significantly stimulate PBMCs to produce large amounts of perforin and granzyme B (*p* < 0.05). Meanwhile, no, or barely detectable, productions of these two factors were found in the untreated PBMC culture supernatants. Notably, ST-4 was more effective than SEC2 with regard to the induction of perforin production (*p* < 0.05), but there was no significant (NS) difference in granzyme B production. These results suggest that SEC2/ST-4 could induce PBMC activation, and release perforin and granzymes to kill the target cells.

### 2.2. Antitumor Activity of SEC2/ST-4 in Ovarian Cancer Cells

To examine the antitumor immune activity of SEC2/ST-4 in ovarian cancer cells, we evaluated the growth inhibition of SKOV3 and ES-2cells. As shown in Figure 2A, both SEC2 and ST-4 showed significant antitumor activities in SKOV3/ES-2 cells (*p* < 0.05). At the same concentration, the antitumor activity of SEC2/ST-4 in SKOV3 cells was significantly higher than that in ES-2 cells (*p* < 0.05). The inhibition rate of ST-4 in SKOV3 cells was significantly higher than that of SEC2 (*p* < 0.05), but the antitumor activity of SEC2/ST-4 was not significant in ES-2 cells. These results suggest that ES-2 cells could trigger a resistance to SEC2/ST-4-induced antitumor immune effects through an unknown mechanism.

### 2.3. Transcriptome Analysis after Treatment in SEC2/ST-4 Condition

The DEGs between SKOV3 and ES-2 cells were identified by RNA-seq after SEC2/ST-4 treatment. As shown in Figure 2B, 291/653 genes were upregulated and 49/155 genes were downregulated after being co-incubated with SEC2/ST-4-treated PBMCs, respectively, in ES-2 cells, while 868/917 genes were upregulated and 895/897 genes were downregulated after being co-incubated with SEC2/ST-4-treated PBMCs, respectively, in SKOV3 cells. The number of DEGs in SKOV3 cell groups was significantly higher than that in ES-2 cell groups after SEC2/ST-4 treatment. Go analysis revealed that these DEGs were highly relevant to the regulation of natural-killer-cell- and T-cell-mediated cytotoxic pathways (Figure 2C–F). Compared with SKOV3 cells, ES-2 cells could resist the SEC2/ST-4-induced antitumor response by interference with the immune-mediated cytotoxicity. Next, we investigated the apoptosis-related DEGs induced by cytotoxic granules in ES-2/SKOV3 cells after treatment with SEC2/ST-4. Among DEGs in the apoptosis signaling pathway, most of the upregulated genes belong to pro-apoptosis genes, such as caspase 8 (CASP8) and DNA damage-inducible transcript 3 (DDIT3), and most of the downregulated genes belong to cell cycle genes, such as check point kinase 1 (CHEK1), cyclin-dependent kinase 1 (CDK1), cyclin B1 (CCNB1) and growth arrest and DNA damage-inducible α (GADD45A) (Figure 2G). The transcriptional level of apoptosis-associated genes in SKOV3 cells was significantly higher than that in ES-2 cells after treatment with SEC2/ST-4. The results suggest that SKOV3 cells were more sensitive than ES-2 cells to cytotoxic granule-mediated apoptosis. As shown in Figure 2H, the transwell co-culture models of transcriptome experiment. (PBMCs were treated with SEC2/ST-4 in the upper chamber and tumor cells in bottom chamber. The cytotoxic particles released by SEC2/ST-4-treated PBMC cells can penetrate the membrane and act on the tumor target cells in the bottom chamber. The antitumor activity of SEC2/ST-4 in SKOV3 cells was significantly higher than that in ES-2 cells.)

### 2.4. Analysis of Cell Stiffness and Regulatory Molecules in Ovarian Cancer Cells

As a monomeric pore-forming protein, perforin-mediated pore formation is also affected by physical and mechanical force [19]. The function of cytotoxic granules is closely related to the stiffness of tumor cells. It was reported that the actomyosin cytoskeleton affected tumor immune evasion, and the softness of cells could prevent CTLs from killing tumors [20]. Cell stiffness is determined by the structure of the cytoskeleton [21]. CDC42 GTPases regulate actin cytoskeleton organization and stiffness for actomyosin contractility by inducing PAK and myosin II activation [22,23]. Among DEGs in the cytoskeleton signaling pathway, we found that the transcriptional level of CDC42EP1, CDC42EP3, RhoGEFs (DOCKs) and PAK3 genes regulated by CDC42 were higher in naïve ES-2 cells than that in naïve SKOV3 cells [16,17,24] (Figure 3A). These results suggest that compared with SKOV3, ES-2 might rely on enhancing CDC42 signal to regulate cell softness.

Actomyosin stress fibers, consisting of F-actin filaments and myosin, are believed to have the most significant contribution to the modulation of cell stiffness and internal tension [25]. To determine whether there are differences in cell stiffness between SKOV3 and ES-2 cells, we detected the structures of F-actin. As shown in Figure 3B, the F-actin filaments in SKOV3 cells form a tangled network with most filaments lying in the peripheral region of the cell. F-actin in ES-2 cells maintains a simple ring-like structure with a lower density. Previously, studies showed that higher absolute levels of F-actin are associated with increased cell stiffness, whereas reductive or low-density F-actin correlates with lower cell stiffness [26,27,28]. Meanwhile, the transcriptional levels of actinin alpha 3 (ACTN3) and Myosin 1A (MYO1A) genes, which increase the cell stiffness, were also significantly lower in naïve ES-2 cells than that in naïve SKOV3 cells [29,30] (Figure 3A). All these results suggest that the stiffness of the ES-2 cell was likely to be lower than SKOV3. Therefore, we next used AFM to measure the stiffness of ES-2 and SKOV3 cells. The Young′s moduli of ES-2 and SKOV3 cells are displayed in Figure 3C,D. In each experiment, 500–1000 original force curves were obtained from 4 to 6 cells, and Young′s moduli were obtained from the Gaussian fitting of Young′s modulus distribution [31]. The Young′s modulus value of ES-2 was approximately 8.32 ± 1.4 kPa, which was significantly lower than that of SKOV3 13.54 ± 0.8 kPa (Figure 3B, *p* < 0.05). The results indicate that the membrane stiffness of the ES-2 cell was significantly softer than that of the SKOV3 cell.

Biophysical studies have revealed that myosin II regulated membrane tension, and increased membrane tension promotes CTL-mediated killing [8,32]. To investigate the role of myosin II in SKOV3/ES-2 cell stiffness, we used blebbistatin, a myosin-II-specific inhibitor, to block myosin activity. As shown in Figure 3E–H, the Young′s modulus values of SKOV3 and ES-2 were significantly increased after pre-treatment with blebbistatin. The results of the cell stiffness measurements demonstrate approximately a two-fold and five-fold increase (compared to untreated group) after treatment with blebbistatin (100 μM) in SKOV3 and ES-2 cells, respectively. These results suggest that activation of myosin II signal involves the regulation of the stiffness in ovarian cancer cells (Figure 3I).

### 2.5. Importance of CDC42 Signaling for SEC2/ST-4-Induced Antitumor Activity

To investigate upstream regulators of myosin II, we first examined a possible involvement of CDC42 signaling. CDC42 regulates specific cytoskeletal events, including actin polymerization and actomyosin assembly [33]. As shown in Figure 4A,B, the protein phosphorylation levels of CDC42, PAK, and MLC2 in ES-2 cells were significantly higher than those in SKOV3 cells. In addition, these protein phosphorylation levels in SKOV3 and ES-2 cells were significantly decreased by CDC42-specific siRNA transfection (*p* < 0.05) (Figure 4C–F). Furthermore, the inhibition properties of SEC2/ST-4 in SKOV3 cells caused an increase of 40% and 33%, respectively, after transfection with CDC42-specific siRNA (Figure 4G). Meanwhile, the inhibition properties of SEC2 and ST-4 in ES-2 cells rose by more than 112% and 95%, respectively, which showed that CDC42 expression is the key factor for SEC2/ST-4 resistance in ES-2 cells. Furthermore, at the same concentration, ST-4 markedly induced enhanced antitumor activity on ES-2 cells compared with the SEC2 group after CDC42-specific siRNA transfection (Figure 4H). These results demonstrate that CDC42 signaling is important for activation of myosin II in ovarian cancer cells. ES-2 cells could rely on enhancing CDC42/MLC2 signal to regulate membrane tension and prevent SEC2/ST-4-induced antitumor immune effect.

### 2.6. Increased Cell Stiffness Improves SEC2/ST-4-Induced Antitumor Activity

To investigate whether SEC2/ST-4-induced antitumor activity can be improved by enhancing the cell stiffness, we used blebbistatin to inhibit myosin II and increase cell membrane tension (Figure 3E,F). We found that the inhibition properties of SEC2 and ST-4 in blebbistatin-treated (100, 50 μM) SKOV3 and ES-2 cells were significantly higher than that of untreated groups, and the inhibition rates had a dose-dependent manner (Figure 5A,B, *p* < 0.05). Furthermore, to assess whether the antitumor effect induced by SEC2/ST-4 is related to the perforin activity after enhancing the tumor cell stiffness, we used a perforin neutralizing antibody to block perforin function. As shown in Figure 5A,B, the blebbistatin-treated stiffer SKOV3/ES-2 cells were significantly rescued in the presence of neutralizing perforin monoclonal antibody after the treatment of SEC2/ST-4 (*p* < 0.05). Taken together, these results suggest that inhibition of the myosin activity increased cell stiffness in ovarian cancer cells. The increased membrane tension is more conductive to perforin-mediated killing.

### 2.7. SEC2/ST-4-Induced S Phase Arresting and Apoptosis in Blebbistatin-Treated Stiffer Ovarian Cancer Cells

We analyzed the cell cycles distribution of blebbistatin-treated stiffer SKOV3/ES-2 cells that were co-incubated in transwell plates with PBMCs in the presence of SEC2/ST-4. As illustrated in Figure 5C–F, the blebbistatin-treated stiffer SKOV3/ES-2 cells presented with SEC2 and ST-4 groups showed enhanced S-phase arrest compared with control groups (*p* < 0.05). Furthermore, adding perforin-neutralizing antibody significantly rescued S- phase arrest of blebbistatin-treated stiffer cells (*p* < 0.05).

Annexin V/PI staining and flow cytometry assay were performed to evaluate the apoptotic levels in blebbistatin-treated stiffer ovarian cancer cells. As shown in Figure 5 G–I, both SEC2 and ST-4 could significantly induce apoptosis in blebbistatin-treated stiffer SKOV3/ES-2 cells (*p* < 0.05). The percentages of both early apoptotic and late apoptotic cells in SEC2 and ST-4 treatment groups were significantly higher than those in the control groups (*p* < 0.05). In addition, the added neutralizing perforin monoclonal antibody groups significantly prevented SEC2/ST-4-induced apoptosis. The results of enhanced early and late apoptosis induced by SEC2 and ST-4 were in accordance with enhanced tumor cell growth inhibition. We speculated that blebbistatin-treated stiffer SKOV3/ES-2 cells’ cell cycle arrest and apoptosis induced by SEC2 and ST-4 contributed to tumor cell growth inhibition.

### 2.8. SEC2/ST-4-Induced Blebbistatin-Treated Stiffer Ovarian Cancer Cells Apoptosis through Caspase Signaling

Caspases cascade activation plays a central role in perforin/granzyme B-induced apoptosis in tumor cells. To further define the tumor cell apoptosis mechanism induced by SEC2/ST-4, we investigated the activities of caspases in blebbistatin-treated stiffer SKOV3/ES-2 cells. As shown in Figure 6A–H, the activated caspase-8, caspase-9, and caspase-3 in SEC2 and ST-4 treatment groups were significantly higher than those in the control groups (*p* < 0.05). At the same concentration, ST-4-activated caspase markers were significantly higher than those activated by SEC2. In addition, the activities of caspases were significantly inhibited by adding the perforin-neutralizing antibody (*p* < 0.05). These data indicate that the intensified caspase signaling transduction was induced by SEC2 and ST-4, ultimately leading to blebbistatin-treated stiffer SKOV3/ES-2 cell apoptosis.

## 3. Discussion

Immunotherapy emerges as a promising treatment for human malignancies, in which CTLs represent a principal of the T-cell-mediated anti-tumor response through providing host immune protection against cancers [34]. In addition, current evidence suggests that CTLs also play an important role in the prevention of chemoresistance in ovarian cancer [35]. Thus, T-cell-based immunotherapy may provide a promising and novel treatment option for ovarian cancer. As a powerful T-cell activator, superantigen SEC2 can effectively trigger CTL-mediated cancer immunotherapy [36]. To enhance the antitumor activity of SEC2, our laboratory constructed an MHC II–independent mutant of SEC2, named ST-4, that exhibits enhanced immunocyte stimulation and antitumor activity [2,5,37]. However, we also found significant variations in the sensitivity of different tumor cell lines to SEC2/ST-4-induced immune cytotoxicity, even if these tumor cells derived from the same type of tumor. Additionally, the escape mechanism of different tumor cell lines against superantigen-induced immune killing is unclear.

As we showed in this study, ST-4 could promote more powerful T-cell activation and human lymphocyte granule-based cytotoxicity than SEC2. However, the ST-4-mediated immune antitumor effect on ovarian cancer cell lines ES-2 and SKOV3 was significantly different. The inhibition rates of ST-4 on ES-2 cells were significantly lower than those in SKOV3 cells (14% in ES-2 cells vs. 49% in SKOV3 cells). In order to clarify the mechanism of SE resistance in ES-2 cells, we first used transcriptome to analyze the DEGs on the SKOV3 cells and ES-2 cells after they had been cocultured with PBMCs in the presence or absence of SE2/ST-4. The result of the Go analysis implies that, compared with SKOV3 cells, ES-2 cells could resist SEC2/ST-4-induced antitumor immune response by negatively regulating the cytotoxic cell-mediated cytotoxicity and apoptosis process.

Perforin is critical for granule-mediated cytotoxicity in NK cells and CTLs. In this process, perforin multimerizes pores in the plasma membranes to allow granzymes to access the target membrane [38]. Recent studies have shown that increasing target cell tension augmented pore formation by perforin and killing by CTLs [8,34]. These reports strongly imply that ES-2 cells might regulate cell softness to prevent SEC2/ST-4-induced perforin killing. In this study, the fluorescent staining result shows that the F-actin in ES-2 cells maintains a simpler ring-like structure with a lower density than that in SKOV3 cells. The data indicate that the stiffness of ES-2 cells might be lower than SKOV3, and this conjecture was subsequently confirmed by AFM-based indentation measurements in our result. However, the mechanisms of ovarian cancer cells that modulate cell stiffness remain incompletely understood.

So far, myosin II is considered to play a pivotal role in the regulation of membrane tension [11,25]. Myosin II hydrolyzes ATP to generate force through attachment and detachment with F-actin filaments. A high myosin II activity and low density of the cross-linked F-actin filament are associated with a low level of cell stiffness [26,39]. In this study, blocking the myosin II by blebbistatin significantly increased the cell stiffness of ovarian cancer cells, as well as enhanced sensitivity of ES-2 cells to SEC2/ST-4-induced immune cytotoxicity.

Previous studies showed that CDC42 GTPases regulate the stiffness by inducing PAK and myosin II activation [23]. Phosphorylation of MLC2 activates myosin II, which enables it to decrease membrane tension [8,40]. Whether or not myosin II is regulated by CDC42 in ovarian cancer cells has not been investigated. In our study, DEG data show that the transcriptional levels of signaling molecules downstream of CDC42 (CDC42EP1, CDC42EP3, DOCKs and PAK) were higher in naïve ES-2 cells than those in naïve SKOV3 cells. Additionally, Western blot results verified that the protein phosphorylation levels of CDC42, PAK, and MLC2 in ES-2 cells were significantly higher than those in SKOV3 cells. By using CDC42-specific siRNA, we found that phosphorylation levels of these proteins were significantly relieved. Furthermore, SEC2/ST-4-induced immune cytotoxicity to ES-2 cells was significantly increased after transfection with CDC42-specific siRNA. These results demonstrate that CDC42 signaling is important for activation of myosin II in ovarian cancer cells, and ES-2 cells might rely on intense CDC42/MLC2 signaling to regulate membrane tension and prevent SEC2/ST-4-induced perforin function.

Caspase-8 and -9 signals are commonly able to activate caspase-3 so that it consequently leads the cleavage of numerous proteins, which corresponds with DNA fragmentation and cellular apoptotic changes [41]. This study shows that the expression levels of pro-apoptotic protein caspase-8, -9, and -3 in blebbistatin-treated stiffer SKOV3/ES-2 cells were significantly up-regulated after treatment with SEC2/ST-4. At the same concentration, ST-4-mediated caspase protein activations were significantly higher than SEC2-mediated activations. Meanwhile, SEC2/ST-4-indcued apoptotic protein expression could be totally reversed using a neutralizing perforin antibody. These results indicate that SEC2/ST-4 induced blebbistatin-treated stiffer SKOV3 and ES-2 cell apoptosis via perforin-dependent signaling. In this process, ST-4 intensified perforin/granzyme signaling to induce enhancer cell apoptosis more than SEC2 did.

## 4. Materials and Methods

### 4.1. Reagents

Recombinant prototype SEC2 and its mutant ST-4 were expressed in engineered *Escherichia coli* BL21 (DE3) strains and purified as earlier described [7,42]. Human lymphocyte separation medium was purchase from Dakewe Biotech Co., Ltd. (Shenzhen, Guangdong, China). Carboxyfluorescein diacetate succinimidyl ester (CFSE) cell proliferation assay and tracking kit and Cell cycle and apoptosis kit were purchased from Beyotime (Haimen, China). CellTiter 96@ aqueous one solution cell proliferation assay kit was purchased from Promega Corp (Madison, WI, USA). Roswell park memorial institute (RMMI)-1640 and fetal bovine serum (FBS) were purchased from Thermo Fisher Scientific (Waltham, MA, USA). Blebbistatin was purchased from Selleck (Houston, TX, USA). FITC-conjugated phalloidin was purchase from Yeasen Biotech Co., Ltd. (Shanghai, China). FITC annexin V apoptosis detection kit with propidium iodide (PI) was from BioLegend (San Diego, CA, USA). Enzyme-linked immunosorbent assay (ELISA) kits for human perforin and granzyme B were purchased from Abcam (Cambridge, UK). Radio Immunoprecipitation Assay (RIPA) lysis buffer was purchased from Beyotime. Antibodies against phospho-CDC42(Ser71), CDC42, phospho-PAK1 (Ser144)/PAK2(Ser141), and phospho-myosin light chain 2(Ser19) were purchased from Cell signaling technology (Boston, MA, USA). Antibodies against cleaved Caspase 3, cleaved Caspase 8, cleaved Caspase 9, GAPDH, goat anti-rabbit IgG (conjugated to horseradish peroxidase, HRP), and anti-perforin antibody were purchased from Abcam.

### 4.2. Cell Lines, Cell Isolation, and Culture

SKOV3 and ES-2 cell lines were purchased from the Cell Bank at the China Academy of Science (Shanghai, China). Human peripheral blood mononuclear cells (PBMCs) were from six healthy donors by Ficoll-Hypaque cell density centrifugation according the manufacture’s instruction. All cells were maintained in RPMI 1640 with 10% FBS.

### 4.3. CFSE Proliferation Assay

PBMCs were obtained and labeled with CFSE as previously described [43]. The labeled PBMCs were resuspended in RPMI-1640 medium supplemented with 10% FBS at final concentration of 5 × 10^6^ cells/mL. All cells were incubated in 48-well flat-bottomed plates at a density of 3 × 10^6^ cells/well in 0.5 mL of culture medium, and then stimulated with SEC2/ST-4 at a final concentration of 100 ng/mL for 72 h. Untreated PBMCs served as negative control. After incubation, cell division analysis was performed using BD Biosciences LSRFortessa, and data were analyzed with FlowJo V10 software (Tree Star, Ashland, OR, USA). % Divided is the percentage of the cells of the original sample that divided.

### 4.4. Cytokine Assay

Freshly isolated PBMCs were incubated and stimulated with SEC2/ST-4 as indicated above. Cell culture supernatants were collected, and the concentrations of perforin and granzyme B in the supernatants were determined using the ELISA kits following the instructions provided by the manufactures. Absorbance values were measured with a microplate reader at a measurement wavelength of 450 nm and a reference wavelength of 620 nm.

### 4.5. RNA-Sequencing and DEGs Analysis

A total of 1 × 10^6^ PBMCs were plated in transwell chambers in absence or presence of 100 ng/mL of SEC2/ST-4, and 1 × 10^5^ SKOV3/ES-2 cells were seeded into each well. Chamber inserts were placed on top of the wells. The membrane with a 0.4 μm proesize in the transwell chambers prevented both cell–cell contact and cell migration but allowed for the diffusion of soluble factors. The plates were incubated in a humidified atmosphere with 5% CO_2_ at 37 °C for 72 h. Then, ES-2/SKOV3 cells were collected and washed with cold PBS. All assays had three replicates. The samples were treated by Personal Biotechnology Co. (Shanghai, China) for transcriptome sequencing.

The edge R package (http://www.rproject.org, accessed on 24 October 2009) was used to determine the DEGs between different treatment groups [44]. The gene expression levels identified in the RNA-Seq sequence were quantified using Transcripts Per Million reads (TPM) by the value of Log2FC into three groups. Genes with expression levels of Log2FC = 2, Log2FC < 2, and Log2FC > 2 showed no change, down-regulated change, and up-regulated change, respectively. The value of the false discovery rate (FDR) of ≤0.05 and |Log2FC| > 2 was used to screen significant DEGs. Gene ontology (GO) enrichment was performed with DAVID (Database for Annotation, Visualization and Integrated Discovery).

### 4.6. Transfection of CDC42 siRNA

The siRNA sequences for human CDC42 (CDC42-1 siRNA, 5′-gac uca aau uga ucu cag att-3′ and 5′-ucu gag auc aau uug agu ctt-3′; CDC42-2 siRNA, 5′-ccu cua uug aga aac utt-3′ and 5′-agu uuc uca aua gua gag gtt-3′) and control (nonspecific siRNA, 5′-uuc ucc gaa cgu guc agg utt-3′ and 5′-acg uga cac guu cgg aga att-3′) were as reported previously [45,46]. Transient transfection of siRNA was performed with the siRNA-Mate transfection regent (Genepharma Co., Shanghai, China.) according to the manufacture′s instruction.

Briefly, 20 pmol of CDC42 siRNA or control was mixed with 2 μL of siRNA-Mate and incubated in 50 μL of serum-free RPMI 1640 medium for 15 min. Then, the siRNA-Mate-siRNA complex was incubated with SKOV3/ES-2 cells in 6-well plates at a density of 1 × 10^6^ cells/wells for 48 h. After that, medium was removed, and cells were detached, washed and resuspended in RPMI-1640 medium supplemented with 10% FBS until further use.

### 4.7. Antitumor Assay

Antitumor effect was evaluated with an MTS assay kit. Briefly, PBMCs at 1 × 10^6^ cells/well as effector cells and SKOV3/ES-2 tumor cells pre-treated with or without siRNA/blebbistatin as target cells were co-cultured at E:T ratios of 10:1 in 96-well plates [43]. SEC2/ST-4 were added to each well at the final concentration of 100 ng/mL. The plate was incubated in a humidified atmosphere of 5% CO_2_ at 37 °C for 72 h. The blank wells (RPMI 1640 only), unsettled cell control wells (SKOV3/ES-2 cell only), and PBMC-releasing wells (PBMCs and proteins) were used as control. Absorbance value was measured with a microplate reader at a test wavelength of 490 nm. Tumor cell viability (%) was calculated with the following equation: [(Abs value in protein-treated cell well − Abs value in PBMC-releasing well)]/(Abs value in unsettled tumor cells control well − Abs value in blank control wells)] × 100. Tumor growth inhibition (%) was 100 − tumor cell viability.

### 4.8. Immunoflourescence Staining

The labeled F-actin networks of SKOV3 and ES-2 cell lines was imaged using fluorescence microscopy, and immunofluorescence staining was performed as previously described [9]. Briefly, the cells were fixed with 1 mL 4% formaldehyde in PBS for 15 min and permeated with 0.2% Triton × 100. Then, the cells were stained with 200 nM FITC-conjugated phalloidin (green) for 30 min. All steps were conducted in a dark room at room temperature. The images of SKOV3/ES-2 cells were taken using a Leika fluorescence microscope (Leika, Wetzlar, Germany).

### 4.9. Measurement of Single Cell Stiffness

An Agilent 5500 AFM instrument (Agilent Technologies, Chandler, AZ, USA) was used to obtain force–distance curves directly from cells cultured in Petri dishes at room temperature (25 °C), as previously described [31]. The probe (κ = 0.036 N/m) was used to precisely apply a compression force orthogonal to the cell. Young′s modulus of cells was calculated from the force–distance curves using the Hertz model as described in previous studies [47]. The number of cells tested in each condition ranged from 6 to 8, with over 550 force–displacement curves generated per condition.

### 4.10. Transwell Assay

Transwell experiments were performed to determine whether perforin and granzyme B were sufficient to induce SKOV3/ES-2 cell apoptosis. SKOV3 and ES-2 cells (1 × 10^6^ cells/well) were seeded in 6-well plates and treated with blebbistatin at final a concentration of 100 μM for 12 h. Then, the treated cells were rinsed twice using RPMI-1640 medium, and 1 × 10^5^ SKOV3/ES-2 cells in 1 mL were seeded into 12-well plates with anti-perforin antibody at a final concentration of 2 μg/mL to block perforin-induced apoptotic signaling. A total of 2 × 10^6^ PBMC in 0.5 mL were plated in each transwell chamber in presence of 100 ng/mL of SEC2 and ST-4. The plates were incubated in a humidified atmosphere of 5% CO_2_ at 37 °C for 72 h.

### 4.11. Evaluation of Cell Cycle Distribution and Cell Apoptosis by Flow Cytometry

After being co-cultured in transwell for 72 h. PI staining was used to analyze DNA content. SKOV3/ES-2 cells were harvested and fixed with 70% ethanol at 4 °C overnight. After treatment with cell cycle and apoptosis kit, DNA content was analyzed on a flow cytometry (BD Biosciences LSRFortessa, San Jose, CA, USA). The percentage of cells in the G1, S, and G2 phases was analyzed with Modfit 5.0 software.

The percentage of cells undergoing apoptosis was determined by double staining with FITC annexin V apoptosis detection kit with PI (Beyotime Biotech. Inc. Shanghai, China), after being co-cultured in transwell for 72 h. SKOV3/ES-2 cells were collected and resuspended in annexin V binding buffer. A total of 100 μL of cell suspension was transferred into a 1.5 mL tube, and 5 μL of FITC annexin V and 10 μL of PI solution were added to each tube. After incubation for 15 min at 25 °C in the dark, 300 μL of annexin V binding buffer was added to each tube. Flow cytometry was performed using a BD Biosciences LSRFortessa, and data were analyzed with FlowJo software.

### 4.12. Western Blotting

After treatment with siRNA or transwell co-culture as indicated above, a total of 5 × 10^6^ SKOV3/ES-2 cells were collected and lysed in RIPA lysis buffer at 4 °C for 10 min to extract cellular protein. Samples containing equal amounts of protein (20–30 μg) were mixed with 5× Laemmli buffer, boiled, and separated on 12% to 15% SDS-PAGE gels. Samples were then transferred onto polyvinylidene fluoride membranes (Millipore, Billerica, MA, USA). After blocking with 5% skimmed milk, primary antibody incubation was conducted overnight at 4 °C in an appropriate dilution. After washing, the blots further incubated for 1 h with HRP-conjugated secondary antibody. Detection was performed using an enhanced chemiluminescence method. In our experiments, we adopt the following algorithm to evaluate the target protein relative expression level. Control group = [Control (Target protein/Housekeeping protein)/Control (Target protein/Housekeeping protein)] = 1.

### 4.13. Statistical Analysis

All values are given as mean ± standard deviation (SD). Data were analyzed using a two-way analysis of variance (ANOVA) method. The follow-up least significant difference (LSD) test was used for post-hoc comparison to assess differences between groups. Differences with values of *p* < 0.05 were considered to be statistically significant.

## 5. Conclusions

Our results show that ST-4 could activate more powerful human lymphocyte granule-based cytotoxicity and antitumor activity than SEC2. Compared with SKOV3 cells, ES-2 cells could escape SEC2/ST-4-induced immune cytotoxicity by enhancing cell softness through the CDC42/MLC2 pathway. Conversely, after enhancing the stiffness of ES-2 cells by blocking the CDC42/MLC2 pathway, SEC2/ST-4 could exhibit a significant antitumor effect, apoptosis and the S-phase cell cycle arrest through the perforin-dependent apoptotic pathway (Figure 7). The results of this study will be useful for understanding the mechanisms of immunotherapy resistance in tumor cells and provide new insight for superantigen-based tumor immunotherapy.

## Figures and Tables

**Figure 1 ijms-24-11796-f001:**
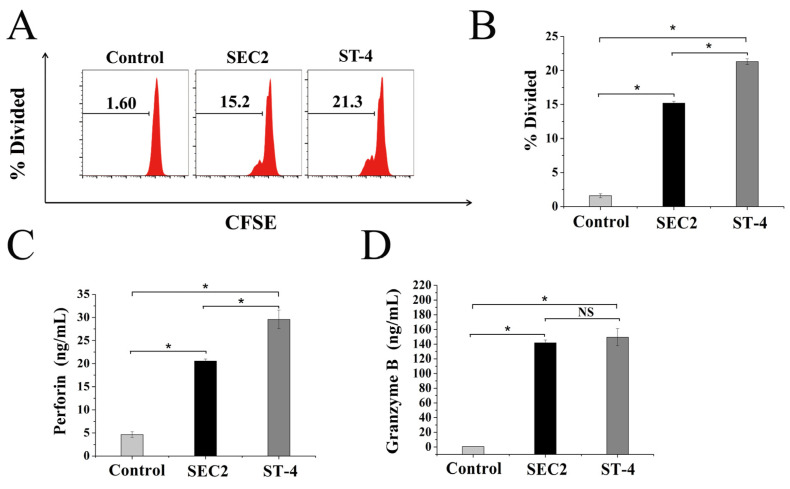
Analysis of SEC2/ST-4-stimulated PBMC proliferation and cytotoxic granule secretion. (**A**,**B**) SEC2/ST-4-stimulated PBMC proliferation. CFSE-labeled PBMCs were stimulated with SEC2/ST-4 at a final concentration of 100 ng/mL. After 72 h, cells were analyzed by flow cytometry. (**C**,**D**) SEC2/ST-4-stimulated cytotoxic granules secretion. PBMCs were stimulated with SEC2/ST-4 at a final concentration of 100 ng/mL. After 72 h, culture supernatants were harvested and used for measurement of perforin and granzyme B by ELISA. Untreated PBMCs served as a negative control. Data are representative of three independent experiments. Each value indicates the mean ± SD of results obtained from three independent experiments. * *p* < 0.05.

**Figure 2 ijms-24-11796-f002:**
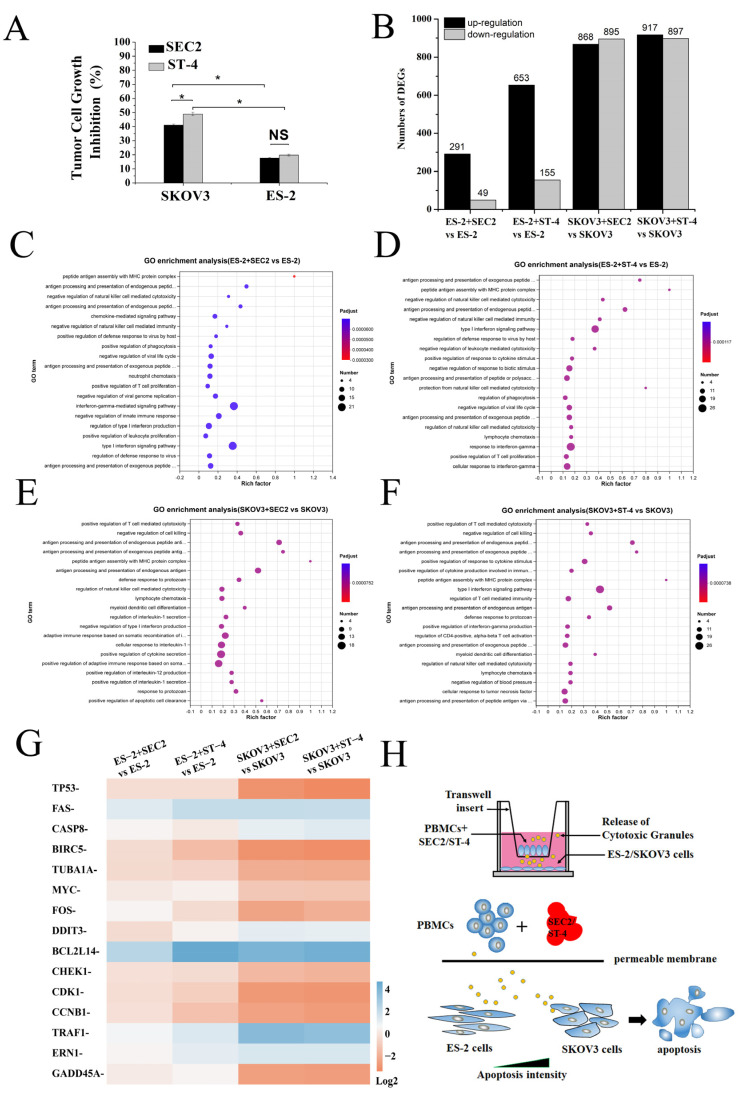
Analysis of apoptosis and transcriptome differences between ES-2 and SKOV3 cells induced by SEC2/ST-4. (**A**) Antitumor activity of SEC2/ST-4 on SKOV3 and ES-2 cells. PBMCs were used as effector cells against SKOV3/ES-2 target cells at E:T ratios of 10:1. The mixed cells were stimulated with SEC2 or ST-4 at 100 ng/mL and incubated for 72 h. The antitumor activity was evaluated with an MTS assay kit. The blank wells (RPMI 1640 only), unsettled cell control wells (SKOV3/ES-2 cell only), and PBMC-releasing wells (PBMCs and proteins) were used as control. Each value indicates the mean ± SD of results obtained from three independent experiments. * *p* < 0.05. (**B**) DEGs identified in ES-2 and SKOV3 cells after being treated with SEC2/ST-4. Histogram shows the number of DEGs compared with the control (ES-2/SKOV3 cells). All assays were performed in three replicates. (**C**–**F**) GO enrichment analysis of DEGs in the treated with SEC2/ST-4 group with the control (ES-2/SKOV3 cells) group. The most significant GO terms were those with corrected *p*-value of <0.05. The rich factor represents the number of DEGs that exist in this term accounting for the total number of genes of this term. (**G**) Heatmap of DEGs in the apoptosis pathway. The vertical axis represents DEGs’ clusters, and the horizontal axis represents the clusters of treated and control groups. (**H**) The transwell co-culture models of transcriptome experiment.

**Figure 3 ijms-24-11796-f003:**
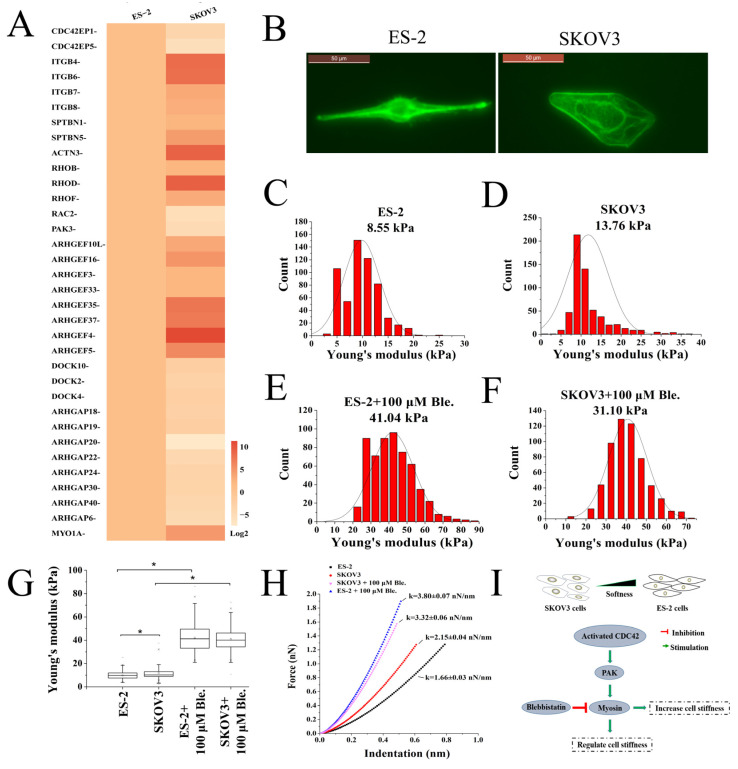
Analysis of the regulation mechanism of cell stiffness in SKOV3 and ES-2 cells. (**A**) Heatmap of DEGs (SKOV3 vs. ES-2 cells) in the cytoskeletal signaling pathway. (**B**) Fluorescence images of F-actin in SKOV3 and ES-2 cells. SKOV3 and ES-2 cells were labeled with FITC-conjugated phalloidin. Then, the images of SKOV3 and ES-2 cells were taken using a Leika fluorescence microscope. (**C**–**F**) Young′s modulus with Gaussian fit function for SKOV3, ES-2 and treated with blebbitatin cells. (**G**) Box chart of modulus values for SKOV3, ES-2 and treated with blebbitatin cells. Horizontal lines in every box representing the average values. Each value indicates the mean ± SD of results obtained from three independent experiments. * *p* < 0.05. (**H**) Force–distance curves in SKOV3, ES-2 and treated with blebbitatin cells. Force–distance curves directly from cell cultured in Petri dishes at 25 °C by AFM. The number of cells tested in each condition ranged from 6 to 8, with over 550 force–displacement curves generated per condition. (**I**) The regulation mechanism of cell stiffness in SKOV3 and ES-2 cells.

**Figure 4 ijms-24-11796-f004:**
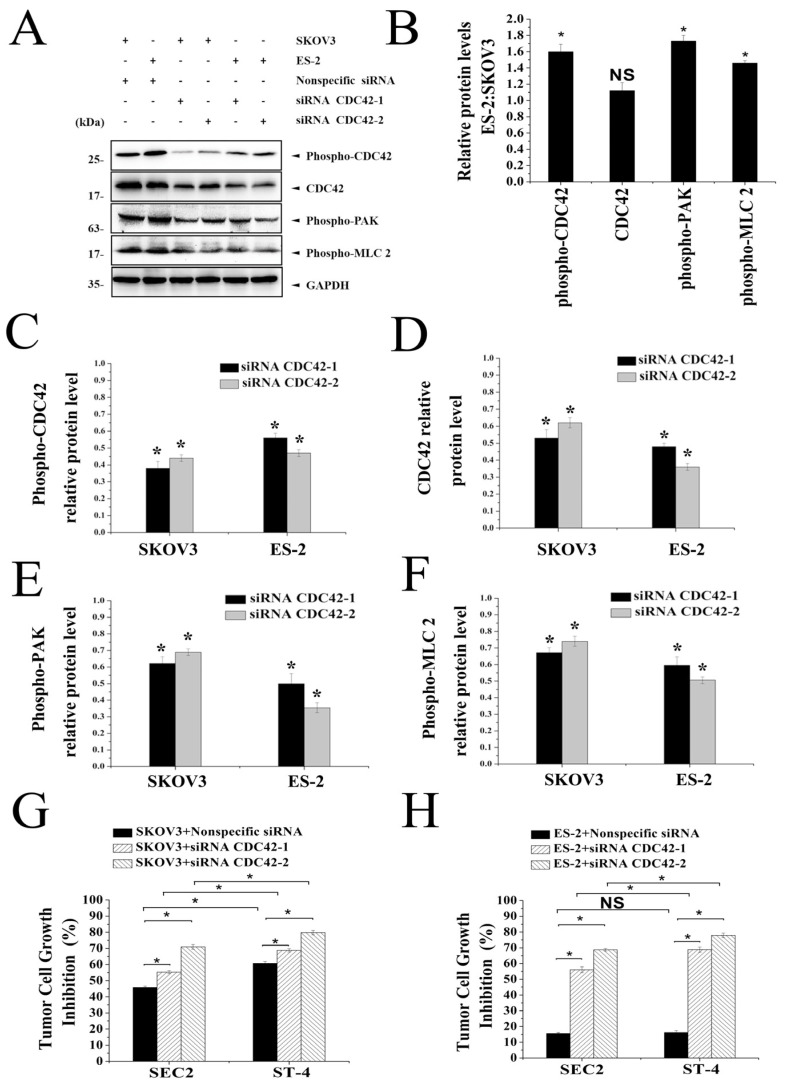
Analysis of CDC42 signaling for SEC2/ST-4-induced antitumor activity. (**A**–**F**) Analysis of the role of CDC42/MLC2 signal in SKOV3 and ES-2 cells. SKOV3 and ES-2 cells were transfected with CDC42 siRNA or nonspecific control siRNA for 48 h. Cells were collected and lysed in RIPA lysis buffer to extract cellular protein. Then, phospho-CDC42, PAK and MLC2 protein levels were detected by Western blot analysis. (**G**,**H**) Antitumor activity of SEC2/ST-4 in SKOV3 and ES-2 cells after transfection with CDC42-specific siRNA. PBMCs were used as effector cells against SKOV3/ES-2 target cells (pre-treated with siRNA CDC42) at E:T ratios of 10:1. The mixed cells were stimulated with SEC2 or ST-4 at 100 ng/mL and incubated for 72 h. The antitumor activity was evaluated with an MTS assay kit. Data are representative of three independent experiments. Each value indicates the mean ± SD of results obtained from three independent experiments. * *p* < 0.05 compared with the control of nonspecific control siRNA.

**Figure 5 ijms-24-11796-f005:**
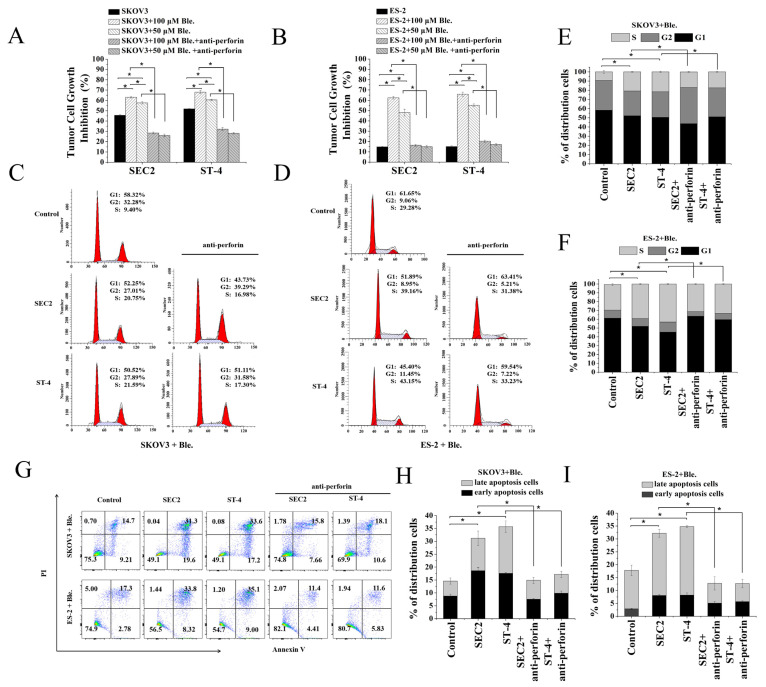
Analysis of SEC2/ST-4-induced antitumor activity in blebbistatin-treated stiffer ovarian cancer cells. (**A**,**B**) Cytotoxicity on tumor cells induced by SEC2/ST-4. PBMCs were used as effector cells. The pre-treated with blebbistatin (100 μM, 50 μM) stiffer SKOV3 and ES-2 cells were used as target cells at E:T ratios of 10:1. The mixed cells were stimulated with SEC2 or ST-4 at 100 ng/mL, and presented with anti-perforin antibody at a final concentration of 2 μg/mL, as indicated in the figure. After 72 h, the antitumor activity was evaluated with an MTS assay kit. The blank wells (RPMI 1640 only), unsettled cell control wells (blebbistatin-treated stiffer SKOV3/ES-2 cell only), and PBMC-releasing wells (PBMCs and proteins) were used as control. (**C**–**F**) S-phase arrest of blebbistatin-treated stiffer SKOV3 and ES-2 cells induced by SEC2 or ST-4 determined by the flow cytometry. After being co-cultured in transwell for 72 h, blebbistatin-treated stiffer SKOV3 and ES-2 cells were harvested and fixed with 70% ethanol at 4 °C overnight and then stained with PI. After treatment, cell cycle was analyzed by flow cytometry. (**G**–**I**) Flow cytometric analysis of Annexin V-FITC/PI-stained cells. The co-cultured cells of PBMC effector and blebbistatin-treated stiffer SKOV3 and ES-2 cells (E:T ratios of 10:1) were treated with SEC2 or ST-4 at 100 ng/mL and presented with anti-perforin antibody as indicated in the figure. After 72 h, blebbistatin-treated stiffer SKOV3 and ES-2 cells had been harvested and stained for analysis. Dot plots of total events are shown with frequencies of cells in each quadrant. Each value indicates the mean ± SD of results obtained from three independent experiments. * *p* < 0.05.

**Figure 6 ijms-24-11796-f006:**
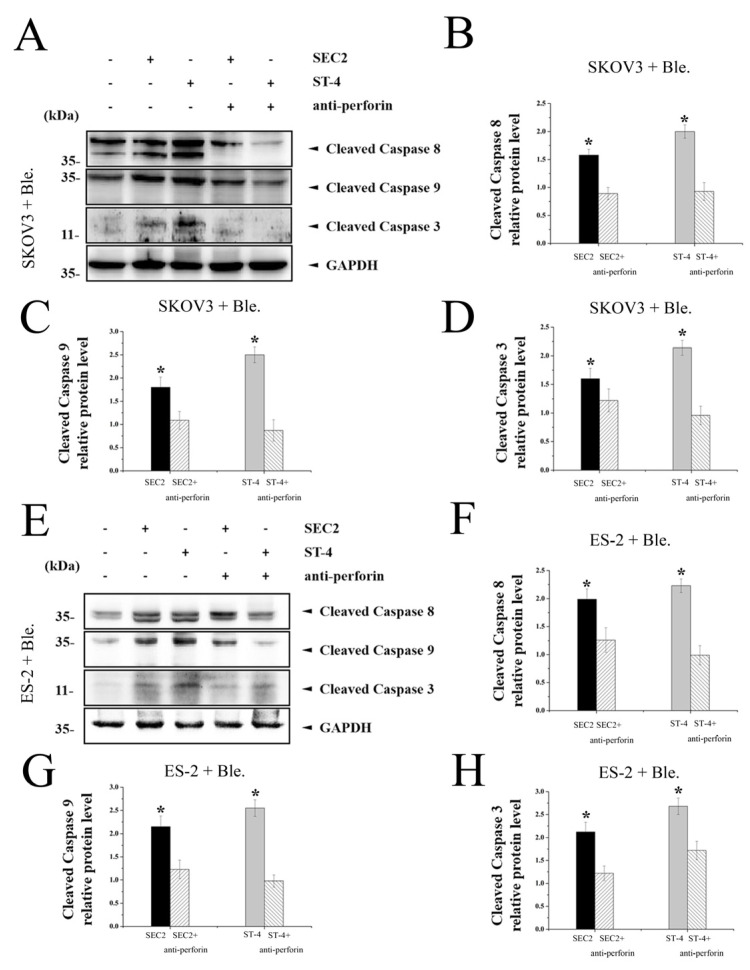
Analysis of SEC2/ST-4-induced apoptotic pathway in blebbistatin-treated stiffer ovarian cancer cells. (**A**–**H**) SEC2/ST-4 treatment of blebbistatin-treated stiffer SKOV3 and ES-2 cells activate apoptosis-related protein caspase 3, 8 and 9. After being co-cultured in transwell for 72 h. Blebbistatin-treated stiffer SKOV3 and ES-2 cells were harvested and lysed in RIPA lysis buffer at 4 °C for 10 min to extract cellular protein. Cleaved caspase 3, 8, and 9 protein levels were detected by Western blot analysis. Each value indicates the mean ± SD of results obtained from three independent experiments. * *p* < 0.05.

**Figure 7 ijms-24-11796-f007:**
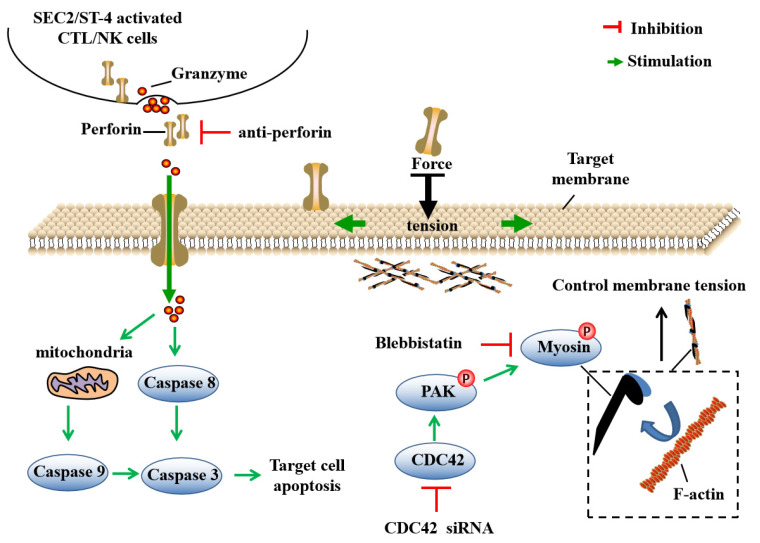
Cell softness prevents SEC2/ST-4-induced cytotoxicity in ovarian cancer cells.

## Data Availability

The transcriptome sequences of ES-2 and SKOV3 cells after treatment with or without SEC2/ST-4 were deposited in NCBI under the accession number PRJNA979936.

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
