# Peer review of "Staphylococcal Enterotoxin C2 Mutant-Induced Antitumor Immune Response Is Controlled by CDC42/MLC2-Mediated Tumor Cell Stiffness"

_ijms, 2023, doi:10.3390/ijms241411796_

Round 1
Reviewer 1 Report
Major Comments:
1. In Figures 1A and B, it would be valuable to label the PBMCs with specific antibodies (CD3 for T cells, CD14 for monocytes, CD19 for B cells, and CD56 for NK cells) to identify the population of PBMCs induced by SEC2/ST-4 for proliferation.
2. In Figure 2A, please include data from the control wells, including PBMCs with SKOV3 and ES-2 without SEC2/ST-4, SKOV3 and ES-2 alone, and PBMCs alone.
3. In Figures 4G-H, it would be valuable to test the synergistic effect of CDC42-1 and CDC42-2 siRNA together.
4. Please provide an explanation in the discussion section of the manuscript for the authors' choice of using PBMCs instead of isolated T cells for the studies, considering that SEC2/ST-4 is a T-cell activator.
5. In line 388, the authors mention that PBMCs were isolated from six healthy donors. Please clarify whether the PBMCs from the healthy donors were pooled and used in all experiments or if PBMCs from different donors were used in separate experiments. If the latter was followed, please explain how the authors accounted for donor variation.
Minor Comments:
1. Please verify the color key (Log2 values) for the heat map in Figure 2G.
2. In Figure 3H, please revise the figure key, as the curves are difficult to read due to the small size of the shapes used in the key.
3. In lines 255/284, for improved clarity, please replace the phrase "stiffer SKOV3/ES-2 cells" with "blebbistatin-treated stiffer SKOV3/ES-2 cells". Please make this change throughout the manuscript.
1. Please proofread the manuscript to correct spelling and spacing errors. Some examples are listed below:
a. In line 56, please correct "To data" to "To date."
b. In line 79, please correct the word "exhibite" to "exhibit."
c. In line 124, please correct the word "anlysis" to "analysis."
d. In line 128, please correct the word "apotosis" to "apoptosis."
e. In line 338, please correct the word "belbbistatin" to "blebbistatin."
Reviewer 2 Report
Fu et al., have done an excellent piece of work that describes a study conducted on the superantigen staphylococcal enterotoxin C2 (SEC2) and its mutant variant, ST-4, which have potential applications in cancer therapy. The researchers aimed to investigate the reasons behind the varying levels of sensitivity to SEC2/ST-4 in different tumor cells and the mechanisms underlying immune resistance in cancer cells.
Fu et al., discovered that ST-4, compared to SEC2, exhibited stronger activation of cytotoxic responses in human lymphocytes. They employed RNA-seq analysis and atomic force microscopy (AFM) to analyze the cells involved.
The findings revealed that certain cancer cells, specifically those with a softer structure (e.g., ES-2 cells), were capable of evading cytotoxic T cell-mediated apoptosis triggered by SEC2/ST-4. This evasion was attributed to the regulation of cell softness through the CDC42/MLC2 pathway.
Conversely, when the stiffness of cancer cells was increased using blebbistatin inhibitor of nonmuscle myosin II, SEC2/ST-4 demonstrated a notable anti-tumor effect on ES-2 cells. This effect was achieved by promoting perforin-dependent apoptosis (a type of programmed cell death) and causing cell cycle arrest in the S-phase.
In summary, Fu et al., found that ST-4 displayed stronger cytotoxic activation compared to SEC2 in human lymphocytes. They identified that cancer cells with softer structures evaded apoptosis induced by SEC2/ST-4 by regulating their softness through the CDC42/MLC2 pathway. However, increasing cell stiffness with a specific inhibitor restored the anti-tumor effect of SEC2/ST-4 by promoting apoptosis and arresting the cell cycle.
I kindly request the authors to address the following questions
1. a spelling mistake for "tension" on line 60, paragraph 3.
2. It would be intriguing to observe the cytokine levels, specifically TNF-alpha and interferon-gamma, both with and without treatment in the SKOV3 and ES2 cell lines.
3. Could you please investigate the behavior of normal cells, such as HEK cells, in relation to the antitumor activity of SEC2/ST-4? I kindly request the authors to include experiments conducted on normal cells as well.
4. I would request to provide a transcriptome analysis excel file to observe the DEGs and their corresponding expression levels. It would have been advantageous to include certain genes for validation purposes using qPCR.
5. Line 135-137 requires further elaboration as it lacks clarity.
6. It would be interesting to include immunofluorescence analysis of F-actin in both treated and untreated cells, as well as comparing them with normal cells. Additionally, the inclusion of Western blot analysis would provide valuable insights.
7. In the analysis of cell stiffness and regulatory molecules in ovarian cancer cells, it would be intriguing to demonstrate downstream pathways using Western blotting to assess the specific impact of blebbistatin treatment on myosin II-related activity. Furthermore, immunofluorescence analysis of both treated and untreated cells would be conducted to visualize the effects on F-actin.
8. Importance of CDC42 signaling for SEC2/ST-4-induced antitumor activity - There seem to be discrepancies in the gel picture. It appears that the CDC42-2 treatment did not work, while for CDC42-1, the effect is only 35-40%. I would suggest trying a different SiRNA and repeating the experiments to obtain clearer images.
9. The current Western experiments conducted to analyze the SEC2/ST-4-induced apoptotic pathway in stiffer ovarian cancer cells lack clarity and effectiveness. It would be beneficial and intriguing to enhance the understanding of this pathway by incorporating additional complementary experiments or repeating the existing experiments to obtain more reliable and improved results.
Round 2
Reviewer 1 Report
Thank you for the revised manuscript.